Noise injection into Freeman chain codes

Lukač Luka 1 luka.lukac@um.si
Nerat Andrej 1
Strnad Damjan 1
Kolingerová Ivana 2
Žalik Borut 1
1 Faculty of Electrical Engineering and Computer Science, University of Maribor , Maribor , Slovenia
2 Department of Computer Science and Engineering, University of West Bohemia , Pilsen , Czech Republic
Winkler Robert
Electronic publication date: 2025 Aug 7
Publication date: 2025
Volume: 11
Electronic Location ID: e3084
Received 2025 Feb 27; Accepted 2025 Jul 5
Copyright: © 2025 Lukač et al.
Copyright year: 2025
Copyright holder: Lukač et al.
License: This is an open access article distributed under the terms of the Creative Commons Attribution License, which permits unrestricted use, distribution, reproduction and adaptation in any medium and for any purpose provided that it is properly attributed. For attribution, the original author(s), title, publication source (PeerJ Computer Science) and either DOI or URL of the article must be cited.
License URL: https://creativecommons.org/licenses/by/4.0/

Keywords: Algorithm, Geometric shape, Boundary alteration, Fractal Dimension

Funding: Slovenian Research and Innovation Agency J2-4458 Research Programme P2-0041 Czech Science Foundation 23-04622L This research was funded by the Slovenian Research and Innovation Agency under Research Project J2-4458 and Research Programme P2-0041, and the Czech Science Foundation under Research Project 23-04622L. The funders had no role in study design, data collection and analysis, decision to publish, or preparation of the manuscript.

==============================
This article presents a novel method for direct noise injection into geometric shapes described by eight-or four-directional Freeman chain codes. Noise is applied to randomly selected segments of a chain code sequence using a set of predefined actions. The design of alterations retains topological characteristics of shapes. The method is tested on various shapes, including open, self-intersecting, and simple shapes, among which the latest two may contain holes. Fractal dimension and mean distance from original are utilised in order to analyse the amount of injected noise in sequences of chain codes. The proposed method enables efficient noise injection directly into Freeman chain codes for use in data augmentation and regularization during neural network training.

Introduction

Noise typically represents an unwanted component of a signal, and dedicated denoising techniques have been developed to remove noise from different types of data (Xu et al., 2016; Ali, El-Dahshan & Yahia, 2017). On the other hand, there are also important applications for deliberate injection of noise into data, such as adding noise into training data of artificial neural networks. Noise injection is an established data augmentation and regularisation concept at artificial neural networks used to avoid occurrences of local minimum phenomena during the training phase (Wang & Principe, 1999) and to prevent overfitting when dealing with digitised shapes (Yin et al., 2015; Jothimani & Premalatha, 2022), and effectively increasing their robustness (Sharma, Grau & Fritz, 2016). Furthermore, some state-of-the-art methods also inject noise into internal components of neural networks in order to enhance their performance (Duan, Ren & Duan, 2022; Chen et al., 2022; Zhang et al., 2023).

Chain coding is a simple and efficient method for describing digitised geometric shapes and binary images (Strnad et al., 2024). Its usage can be found in industry (Honi et al., 2022), medicine (Aldemir et al., 2023; Fanti et al., 2025), computer vision (Althobaiti & Lu, 2017; Boukharouba & Bennia, 2017; Zhao & Liu, 2020; Wang & Wang, 2022), and efficient representations of tree models (Strnad et al., 2019). Furthermore, despite the fact that chain codes already provide a compact and storage-efficient representation of shapes (Sánchez-Cruz, Bribiesca & Rodríguez-Dagnino, 2007; Tapia-Dueñas, López & Sánchez-Cruz, 2025), they can be compressed even further using entropy encoding methods (Žalik et al., 2018, 2021). Particular efficiency of such techniques was achieved in cartoon (Jeromel & Žalik, 2020) and binary image compression (Zahir, Dhou & George, 2007).

Injecting noise into chain codes without the need to decode (i.e., rasterise) and re-encode them would significantly improve the efficiency of the process, and enable better control over the preservation of geometric shapes’ topological characteristics. If noise is injected into decoded raster images, there is no control over topological characteristics of geometric shapes. In the case of machine learning (ML), the changes of topological characteristics can significantly impact the classification accuracy. The main advantages of the proposed approach are the preservation of topological characteristics of geometric shapes and noise injection efficiency. To the best of our knowledge, there exist no methods for injecting noise directly into chain codes. In this article, we propose the first method for direct injection of noise into geometric shapes described by Freeman chain codes. The main limitation of the proposed method is its limitation to the Freeman chain codes as other chain code types would require different sets of noise injection rules.

Chain codes represent the shape’s boundary by a sequence of consecutive directional commands. Various types of chain codes have been developed (Bribiesca, 1999; Sánchez-Cruz & Rodríguez-Dagnino, 2005; Kabir, 2015; Žalik et al., 2016). The most recent chain coding approaches include agent-based modelling and biological computing for encoding binary images (Dhou & Cruzen, 2019; Dhou, 2020; Dhou & Cruzen, 2021), and multi-resolution chain codes (Nerat et al., 2024; Žalik, 2024). However, the pioneer chain codes were proposed as far back as 1961 (Freeman, 1961). Fundamentally, two versions of Freeman chain codes were introduced. The first one is the eight-directional chain code (F8), where the shape boundary is described using symbols from the alphabet ∑F8={0,1,2,3,4,5,6,7}. The angle of movement for each symbol c∈ΣF8 in the raster space (according to the x-axis) is calculated as ϕ=45∘×c (see Fig. 1A). The four-directional Freeman chain code (F4) uses the same principles. It, however, consists of a smaller alphabet containing just four symbols: ∑F4={0,1,2,3}. Consequently, the angle of movement for symbol c∈∑F4 is calculated as ϕ=90∘×c. Possible movements of the F4 chain code are shown in Fig. 1B.

Figure 1 Freeman chain code symbols: (A) F8, (B) F4.

While the F8 chain code traverses through centres of boundary pixels (see Fig. 2A), the F4 chain code can be used in two ways. The first option moves along the centres of boundary pixels (Fig. 2B), while the second operates along boundary pixels’ edges. It is often referred to as a crack code (Wilson, 1997) (see Fig. 2C).

Figure 2 Encoding of a shape by Freeman chain codes: (A) F8, (B) F4 (centre), (C) F4 (crack).

The start of each chain code sequence is marked by a filled circle.

Closed shapes can contain holes, therefore, they are described by one or more chain code sequences. The outer boundary of the shape is often referred to as a loop, while the other, inner boundaries, are considered rings (Mortenson, 1997). An open shape, however, consists of only one chain, also known as a path. The proposed method is not limited in regards to characteristics of shapes. It works with either open or closed, self-intersecting, or non-self-intersecting shapes, while preserving their original topological characteristics (number of rings and number of self-intersections).

The main contributions of the article are: The first method for noise injection directly into eight-and four-directional Freeman chain codes, preserving the topological characteristics of chain code sequences.

The design of alteration rulesets for both types of Freeman chain codes, which define noise injection with local perturbations of chain code sequences.

Experimental work, which demonstrates the efficacy of the proposed method on a set of open, self-intersecting, and simple geometric shapes that may contain holes, and the analysis of the injected noise with fractal dimension and mean distance from original (Rogelj, Hudej & Petric, 2013).

Method

Let S be a geometric shape described by a set of chains C={Ci};0≤i<|C|. |C| denotes the number of chain code sequences, required to describe the boundary of S. If S is a closed shape, C0∈C represents a loop, while Ci∈C;1≤i≤|C| represent rings (i.e., holes). Otherwise, if S is an open shape, it consists of only C0∈C, which represents a path. Each chain consists of a sequence of chain code symbols Ci=⟨ci,j⟩;ci,j∈∑{F8/F4};1≤j≤|Ci|. If an F4 chain code sequence is given as a crack code sequence, in our case, it is first converted to a format that encodes movements between centres of boundary pixels. The block diagram that outlines the sequence of steps during noise injection is shown in Fig. 3.

Figure 3 Block diagram of the noise injection method.

Noise is introduced separately to each Ci, where randomly chosen chain codes are injected with noise according to a set of predefined rules. The rules ensure that a loop or a ring is not converted to a path, and vice versa. An example of noise injection to an F8 chain code sequence is shown in Fig. 4.

Figure 4 Noise injection into the boundary of the shape denoted with an F8 loop: (A) before the noise injection (the red arrows indicate a pair of chain codes where noise shall be injected), (B) after the noise injection (the red arrows indicate the altered pairs of chain codes).

The noise injection method for a geometric shape described with Freeman chain code sequences is explained in Algorithm 1. As input, the algorithm receives a set of all chains C, type of the provided chain code sequences, probability of an alteration μ (an arbitrary value from the interval (0,1]), and a number of noise injection iterations n (an arbitrary positive integer value). μ and n are user-defined parameters. The loop in line 9 iterates through all chain code symbols ci,j∈Ci. In line 12, the altered chain code sequence section A is obtained according to randomly selected chain code symbol ci,j and its descendant ci,j+1. The alteration is performed according to replacement rules defined later in this section. Afterwards, if an alteration does not change the amount of self-intersections in the chain Ci or the amount of intersections with another chain Cj∈C, ci,j and ci,j+1 are replaced by A in line 14. The set C of chain code sequences with injected noise is obtained as the result of the algorithm after the function is repeated for all chain code sequences for n times.

Algorithm 1 Noise injection into S described by Freeman chain code sequences.

1: function noise-injection ( C, type, μ, n)	
2:                       ⊳C: chain code sequences	
3:                           ⊳type: {F8/F4}	
4:                 ⊳μ∈(0,1]: probability of an alteration	
5:                       ⊳n>0: number of iterations	
6: ⊳ Returns: geometric shape with noisy chain code sequences	
7:  for l←1ton do	
8:   for i←1to|C| do	
9:    for j←1to|Ci|−1 do	
10:     r← Random(0, 1)	
11:     if r<μ then	
12:      A← Alteration(ci,j, ci,j+1, type)	
13:      if PreservedTopologicalCharacteristics( C, A) then	
14:       Ci = InjectNoise(j, Ci, A)	
15:      end if	
16:     end if	
17:    end for	
18:   end for	
19:  end for	
20:	
21:  return C	
22: end function	

Noise injection method operates using the same principles for both types of Freeman chain code sequences. The sole difference lies in alteration type selection, where separate alteration rulesets have been designed for F8 and F4 chain codes according to their specifics. The rulesets for both types of alterations were designed in a way that retains the property of openness of a chain code sequence after the noise is injected.

Alteration options for F8

There are seven different alteration options when injecting noise into F8 chain code sequences: REMOVE,

PUSH,

WRAP,

FLIP,

PULL,

CUT,

ROTATE.

The alterations are selected based on ci,j and ci,j+1 and are formally given in Eq. (1). In the right column, there are various conditions that define the selection of the appropriate alteration in the left column.

(1) f(ci,j,ci,j+1)={fREMOVE();|ci,j−ci,j+1|=4fPUSH(ci,j);(ci,j=ci,j+1)∧(ci,jmod2=0)fWRAP(ci,j);(ci,j=ci,j+1)∧(ci,jmod2=1)fFLIP(ci,j,ci,j+1);(|ci,j−ci,j+1|=1)∨(|ci,j−ci,j+1|=7)fPULL(ci,j,ci,j+1);((|ci,j−ci,j+1|=2)∨(|ci,j−ci,j+1|=6))∧(ci,jmod2=1)fFLIP/CUT(ci,j,ci,j+1);((|ci,j−ci,j+1|=2)∨(|ci,j−ci,j+1|=6))∧(ci,jmod2=0)fROTATE/CUT(ci,j,ci,j+1);(|ci,j−ci,j+1|=3)∨(|ci,j−ci,j+1|=5).

The effects of all types of alterations will be demonstrated using the path in Fig. 5, whose Ci consists of the following chain code symbols: 370011233542441.

Figure 5 A path used for demonstrating the alterations for F8 chain codes.

The start of the path is marked by a filled circle.

REMOVE erases a pair of ci,j and ci,j+1, as displayed in Fig. 6.

(2) fREMOVE()=⟨⟩.

Figure 6 REMOVE applied to a pair of F8 chain code symbols (marked with red arrows).

Example: 370011233542441→0011233542441.

PUSH operates on a pair of equal consecutive chain code symbols, representing a horizonal or a vertical segment. They are transformed in a way that a turn is obtained as given in Eq. (3) and shown in Fig. 7. A turn can be produced in two directions: up and down (horizontal segment), or left and right (vertical segment), chosen on a random basis.

(3) fPUSH(ci,j)={⟨(ci,j+1)mod8,(ci,j+7)mod8⟩⟨(ci,j+7)mod8,(ci,j+1)mod8⟩.

Figure 7 PUSH applied to a pair of F8 chain code symbols (marked with red arrows).

Example: 370011233542441→371711233542441.

WRAP contains two options to apply an alteration to pairs of diagonal steps in the same direction: up and down, which are selected on a random basis. The action is given in Eq. (4) and shown in Fig. 8.

(4) fWRAP(ci,j)={⟨(ci,j+1)mod8,ci,j,(ci,j+7)mod8⟩⟨(ci,j+7)mod8,ci,j,(ci,j+1)mod8⟩.

Figure 8 WRAP applied to a pair of F8 chain code symbols (marked with red arrows).

Example: 370011233542441→3700012233542441.

FLIP performs a transformation where ci,j and ci,j+1 switch their roles, as given in Eq. (5). Consequently, a left turn is transformed into a right one and vice versa as displayed in Fig. 9. (5) fFLIP(ci,j,ci,j+1)=⟨ci,j+1,ci,j⟩.

Figure 9 FLIP applied to a pair of F8 chain code symbols (marked with red arrows).

Example: 370011233542441→370011323542441.

PULL represents an opposite operation to PUSH, as two same consecutive chain code symbols are obtained from a turn. The latter is given in Eq. (6) and displayed in Fig. 10.

(6) fPULL(ci,j,ci,j+1)={⟨ci,j+ci,j+12mod8,ci,j+ci,j+12mod8⟩;|ci,j−ci,j+1|=2⟨(ci,j+ci,j+1)mod82,(ci,j+ci,j+1)mod82⟩;|ci,j−ci,j+1|=6.

Figure 10 PULL applied to a pair of F8 chain code symbols (marked with red arrows).

Example: 370011233542441→370011234442441.

CUT is a type of alteration that transforms ci,j and ci,j+1 into a single connection according to specific conditions (right column) given in Eq. (7) (see Fig. 11).

(7) fCUT(ci,j,ci,j+1)={⟨ci,j+ci,j+12⟩;|ci,j−ci,j+1|=2⟨⌈ci,j+ci,j+12⌉⟩;|ci,j−ci,j+1|=3 ∧((ci,j+ci,j+1=3) ∨(ci,j+ci,j+1=7) ∨(ci,j+ci,j+1=11))⟨⌊ci,j+ci,j+12⌋⟩;|ci,j−ci,j+1|=3 ∧((ci,j+ci,j+1=5) ∨(ci,j+ci,j+1=9))⟨(max{ci,j,ci,j+1}+1)mod8⟩;|ci,j−ci,j+1|≠3 ∧((ci,j+ci,j+1=5) ∨((ci,j+ci,j+1=6) ∨(ci,j+ci,j+1=9))⟨(min{ci,j,ci,j+1}−1)mod8⟩;|ci,j−ci,j+1|≠3 ∧ci,j+ci,j+1=7.

Figure 11 CUT applied to a pair of F8 chain code symbols (marked with red arrows).

Example: 370011233542441→37001123353441.

The final alteration of F8 chain code is ROTATE, which rotates ci,j and ci,j+1 by 45∘. An example of such action is given in Eq. (8) and shown in Fig. 12.

(8) fROTATE(ci,j,ci,j+1)={⟨(ci,j+1)mod8,(ci,j+1+1)mod8⟩;(ci,j+ci,j+1=3)∨(ci,j+ci,j+1=7)∨(ci,j+ci,j+1=11)⟨(ci,j+7)mod8,(ci,j+1+7)mod8⟩;(ci,j+ci,j+1=5)∨(ci,j+ci,j+1=9).

Figure 12 ROTATE applied to a pair of F8 chain code symbols (marked with red arrows).

Example: 370011233542441→370011233542430.

Alteration options for F4

In contrast to F8, F4 has only three possible alterations: WRAP,

REMOVE,

FLIP.

The ruleset of all possible alterations is given in Eq. (9). In the continuation, the alterations are given and displayed on the path shown in Fig. 13.

(9) f(ci,j,ci,j+1)={fWRAP(ci,j);ci,j=ci,j+1fREMOVE();|ci,j−ci,j+1|=2fFLIP(ci,j,ci,j+1);otherwise.

Figure 13 A path used for demonstrating the alterations for F4 chain codes.

The start of the path is marked by a filled circle.

The effects of alterations will be demonstrated using the path in Fig. 13, whose Ci consists of the following chain code symbols: 000113.

WRAP transforms a straight line into a U-turn as per Eq. (10). There are two options to perform this action: up and down (horizontal segment), or left and right (vertical segment), as shown in Fig. 14. The alteration is chosen on a random basis.

(10) fWRAP(ci,j)={⟨(ci,j+1)mod4,ci,j,ci,j,(ci,j+3)mod4⟩⟨(ci,j+3)mod4,ci,j,ci,j,(ci,j+1)mod4⟩

Figure 14 WRAP applied to a pair of F4 chain code symbols (marked with red arrows).

Example: 000113→30010113.

As per its name, REMOVE removes ci,j and ci,j+1 according to Eq. (11) (see Fig. 15).

(11) fREMOVE()=⟨⟩

Figure 15 REMOVE applied to a pair of F4 chain code symbols (marked with red arrows).

Example: 000113→0001.

FLIP flips the roles of ci,j and ci,j+1. As a result, a left turn is transformed into a right turn and vice versa, as shown in Eq. (12) and Fig. 16.

(12) fFLIP(ci,j,ci,j+1)=⟨ci,j+1,ci,j⟩

Figure 16 FLIP applied to a pair of F4 chain code symbols (marked with red arrows).

Example: 000113→001013.

Preservation of topological characteristics

In the final step before accepting an alteration, potential changes of the topological genus in the variated S are checked. The routine verifies whether the replacement of ci,j and ci,j+1 with A would change the number of holes and self-intersections in the observed shape. In that case, such alteration is not accepted. Example of an alteration that changes the topological characteristics of S can be seen in Fig. 17.

Figure 17 S: (A) before the alteration, (B) after the alteration.

The pixel that changes the topological characteristics of S is marked yellow.

In another case, if S already contains self-intersections before an alteration is performed, the same procedure is utilised when injecting noise into S. The topological characteristics of S are preserved in this case as well. Self-intersections that are present in S in the beginning are preserved no matter the amount of noise injection iterations. An example of an alteration that changes the topological genus of a self-intersecting S is shown in Fig. 18.

Figure 18 S: (A) before the alteration, (B) after the alteration.

The pixel that changes the topological characteristics of S is marked yellow while the self-intersection is marked red.

Implementation-wise, chain code sequences Ci∈C are implicitly rasterised before noise injection since the topological characteristics are observed in raster space. The pixels belonging to the rasterised chain code sequences are stored in a hash table using the following hash function h:

(13) h(x,y)=xmax⋅y+x,

where x and y are rasterised pixel coordinates, and xmax is the maximum x coordinate. After an alteration is selected, the procedure checks whether any pixel in the altered chain code sequence borders any other pixel in segments of all chain code sequences apart from its neighbouring pixels. In such case, the alteration is rejected. Otherwise, the alteration is accepted and injected into Ci, while the hash table is updated accordingly.

Time and space complexity

Let L0 represent the total number of chain code symbols of an input geometric shape S, which is calculated using:

(14) L0=∑i=1|C||Ci|.

In each iteration of noise injection, the total number of chain code symbols that describe S can increase/decrease depending on the number of performed alterations. Each REMOVE operation decreases a chain code sequence length by 2, each CUT decreases it by 1, and each WRAP increases it by 1 (F8) or 2 chain code symbols (F4). Let |REMOVE|i, |CUT|i, and |WRAP|i represent the numbers of performed alterations REMOVE, CUT, and WRAP in the i-th noise injection iteration. For F8 chain codes, let ki be defined as:

(15) ki=(|WRAP|i−|CUT|i−2⋅|REMOVE|i)Li.

where Li represents the total number of chain code symbols after i noise injection iterations. Similarly, for F4 chain codes, let ki be defined as:

(16) ki=(2⋅|WRAP|i−2⋅|REMOVE|i)Li.

The total number of chain code symbols after the first iteration in the case of F8 chain codes is calculated as:

L1=L0+(|WRAP|0−|CUT|0−2⋅|REMOVE|0)==L0+k0L0==L0(1+k0).

After the second noise injection iteration, the total number of chain code symbols is calculated as:

L2=L1+(|WRAP|1−|CUT|1−2⋅|REMOVE|1)==L1+k1L1==L0+k0L0+k1(L0+k0L0)==L0(1+k0+k1+k0k1)==L0(1+k0)(1+k1).

Continuing in this manner, the total length of chain codes can be expressed as:

(17) Li=L0∏j=0i−1(1+kj).

For k≪1, this equation can be approximated by:

(18) Li=L0(1+ki¯)i,

where (1+ki¯) is the geometric mean of terms in the product.

The time complexity of the proposed method depends on the total number of chain code symbols of the input geometric shape S. During the iteration through L0 chain code symbols, each selection of a suitable alteration, detection of potential topological characteristic changes as well as local alteration of a chain code sequence are performed in O(1). According to this and the previous derivation of Li, the overall time complexity of the proposed method is T(n)=L0∑i=0n−1(1+ki¯)i. The space complexity follows the same pattern except that it includes the implicit rasterisation (for preservation of topological characteristics). Therefore, it is expressed as S(n)=2⋅L0(1+kn¯)n.

Results and discussion

The proposed method was tested on a set of different geometric shapes, which can be accessed at GeMMA (2025) and represent a selection of shapes with different characteristics (number of chain code symbols, number of loops, etc.). Among those, Line Segment, Square, Twist, Ankh, Camel, and Butterfly were chosen for the demonstration purposes. Each shape is described with both F8 and F4 chain code sequences. Test shapes along with their basic properties (number of chains, total number of chain codes in F8 and F4 encodings) are collected in Table 1 while the shapes are displayed in Fig. 19.

Table 1 Properties of test geometric shapes.

Shape	|C|	Number of chain code symbols (F8)	Number of chain code symbols (F4)	
Line Segment	1	200	200	
Square	1	400	400	
Twist	1	1,600	3,200	
Ankh	2	1,418	1,734	
Camel	10	13,858	20,156	
Butterfly	127	46,172	64,374	

Figure 19 Test geometric shapes: (A) Line Segment, (B) Square, (C) Twist, (D) Ankh, (E) Camel, (F) Butterfly.

Fractal Dimension (FD) (Mandelbrot, 1967) is an index that indicates the ruggedness of a pattern and its detail complexity (Allen, Brown & Miles, 1995). It is given in Eq. (19):

(19) FD=ln⁡N(ε)ln⁡1ε

where N denotes the number of pixels that lie on the shape’s boundary, while ε represents the ratio between the pixel side and the size of geometrical space.

Mean Distance from Original (MDfO) measures the average distance between the pixels of the original and the perturbed geometric shape as (Rogelj, Hudej & Petric, 2013):

(20) MDfO=1|C|∑i=1|C|1|Ci|∑j=1|Ci|minci,k0∈Ci0‖ci,j−ci,k0‖

where Ci0 represents an original chain code sequence without injected noise.

Noise injection into test geometric shapes was evaluated with FD and MDfO. Tables 2 and 3 present the values of FD and MDfO after 100 noise injection iterations with different alteration probabilities μ. As expected, both FD and MDfO increase with higher values of μ. It can be observed that F4 test shapes are affected by noise injection more than their F8 counterparts. This difference can be attributed to the definition of the common alteration WRAP in F4, which increases the number of chain code symbols by 2, whereas the maximum increment of chain code symbols in a single alteration in F8 is only 1.

Table 2 FD and MDfO values after 100 iterations of noise injection into F8 test shapes using different values of μ.

Shape	Metric	Noise injection probability	
μ=0.01	μ=0.05	μ=0.10	μ=0.20	μ=0.50	
Line Segment	FD	1.00	1.01	1.03	1.04	1.11	
MDfO	0.39	0.84	1.01	1.16	2.25	
Square	FD	1.29	1.29	1.31	1.33	1.40	
MDfO	0.47	0.72	0.96	1.43	2.25	
Ankh	FD	1.21	1.23	1.24	1.27	1.32	
MDfO	0.45	0.80	1.96	1.38	2.08	
Twist	FD	1.28	1.31	1.33	1.34	1.40	
MDfO	0.49	0.88	1.12	1.35	2.03	
Camel	FD	1.23	1.25	1.26	1.28	1.33	
MDfO	0.39	0.75	0.99	1.31	2.04	
Butterfly	FD	1.39	1.40	1.41	1.43	1.48	
MDfO	0.44	0.79	1.01	1.34	2.08	

Table 3 FD and MDfO values after 100 iterations of noise injection into F4 test shapes using different values of μ.

Shape	Metric	Noise injection probability	
μ=0.01	μ=0.05	μ=0.10	μ=0.20	μ=0.50	
Line Segment	FD	1.12	1.27	1.36	1.47	1.60	
MDfO	0.96	2.70	3.85	6.89	16.28	
Square	FD	1.40	1.55	1.65	1.73	1.81	
MDfO	0.77	2.31	3.70	6.99	14.66	
Ankh	FD	1.30	1.43	1.53	1.63	1.72	
MDfO	0.58	2.05	3.62	6.09	11.68	
Twist	FD	1.38	1.48	1.53	1.67	1.80	
MDfO	0.59	1.91	3.32	5.98	13.51	
Camel	FD	1.30	1.39	1.47	1.55	1.66	
MDfO	0.57	2.00	3.33	5.40	11.29	
Butterfly	FD	1.45	1.54	1.62	1.69	1.78	
MDfO	0.57	1.98	3.37	5.91	11.00	

The value μ=0.10 was used in the rest of experiments in order to observe gradual effects of noise injection over an increasing number of iterations. Injection of noise into both F8 and F4 shapes is visually demonstrated in Figs. 20–22, where subfigures (A), (B), and (C) indicate F8 shapes, while (D), (E), and (F) indicate F4 shapes.

Figure 20 Line Segment after different numbers of noise injection iterations: (A, D) n = 10, (B, E) n = 50, (C, F) n = 100.

Figure 21 Twist after different numbers of noise injection iterations: (A, D) n = 10, (B, E) n = 50, (C, F) n = 100.

Figure 22 Butterfly after different numbers of noise injection iterations: (A, D) n = 10, (B, E) n = 50, (C, F) n = 100.

Noise injection into Line Segment is shown in Fig. 20. Among the test shapes, it is the only open shape. Therefore, it does not contain any loops or rings as it is described by a path instead. It should be noted that Line Segment remains open no matter the number of noise injection iterations.

Figure 21 represents Twist, a shape that is described by a single loop. Topologically, Twist is different from other test shapes since it contains four self-intersections. It can be seen that the injected noise does not change the shape’s topological characteristics.

The last presented shape is Butterfly (Fig. 22), which consists of a loop and 126 rings. The noise injection method successfully handled the shape despite the fact that it is considerably complicated.

In Tables 4 and 5, FD and MDfO evaluation values are collected at different numbers of noise injection iterations for F8 and F4 geometric shapes, respectively. FD values generally increase as more noise is applied to shapes (Fig. 23). Due to its limited function range, FD tends to converge towards the upper limit of the range (indicating a shape that is essentially dominated by noise). Hence, FD can also serve as a criterion for determining a stop condition for noise injection (e.g., when after several iterations FD rises only marginally). On the other hand, MDfO exhibits a more continuous growth across iterations (Fig. 24). Unlike FD, the initial value of MDfO is zero for all geometric shapes, making it particularly suitable for estimating the amount of injected noise. This property also enables comparisons of noise levels embedded in different geometric shapes. Furthermore, it can be seen that the growth of MDfO slows down beyond a certain number of noise injection iterations, particularly for geometric shapes with many rings. The reason for this is that the raster space in the interior of a geometric shape gets occupied and the geometric shape can be expanded only outwards, causing many alterations to be rejected.

Table 4 FD and MDfO values according to increasing numbers of noise injection iterations into F8 test shapes.

Shape	Metric	Number of iterations	
n=1	n=10	n=50	n=100	n=500	n=1,000	
Line Segment	FD	1.00	1.00	1.01	1.03	1.17	1.30	
MDfO	0.07	0.46	0.71	1.01	2.54	4.35	
Square	FD	1.30	1.29	1.29	1.31	1.44	1.56	
MDfO	0.07	0.40	0.67	0.96	2.80	4.62	
Ankh	FD	1.21	1.21	1.23	1.24	1.37	1.47	
MDfO	0.08	0.49	0.75	0.96	2.51	4.13	
Twist	FD	1.24	1.28	1.31	1.33	1.42	1.52	
MDfO	0.12	0.50	0.87	1.12	2.52	4.36	
Camel	FD	1.23	1.23	1.25	1.26	1.35	1.44	
MDfO	0.09	0.46	0.74	0.99	2.50	4.15	
Butterfly	FD	1.38	1.39	1.40	1.41	1.50	1.58	
MDfO	0.09	0.43	0.78	1.01	2.48	4.17	

Table 5 FD and MDfO values according to increasing numbers of noise injection iterations into F4 test shapes.

Shape	Metric	Number of iterations	
n=1	n=10	n=50	n=100	n=500	n=1,000	
Line Segment	FD	1.03	1.11	1.25	1.36	1.62	1.72	
MDfO	0.23	0.85	2.40	3.85	18.43	38.34	
Square	FD	1.32	1.40	1.55	1.65	1.80	1.83	
MDfO	0.21	0.84	2.32	3.70	16.61	34.29	
Ankh	FD	1.26	1.30	1.43	1.53	1.73	1.77	
MDfO	0.12	0.58	1.99	3.62	12.62	19.85	
Twist	FD	1.35	1.35	1.42	1.53	1.79	1.84	
MDfO	0.12	0.57	1.89	3.32	15.69	31.69	
Camel	FD	1.28	1.30	1.39	1.47	1.68	1.75	
MDfO	0.09	0.57	1.86	3.33	12.68	23.59	
Butterfly	FD	1.43	1.45	1.54	1.62	1.78	1.82	
MDfO	0.10	0.55	1.91	3.37	11.74	13.92	

Figure 23 Fractal dimension (FD) values for the test shapes according to different numbers of noise injection iterations: (A) F8, (B) F4.

Figure 24 Mean distance from original (MDfO) values for the test shapes according to different numbers of noise injection iterations: (A) F8, (B) F4.

Conclusions

This article introduces a method for injection of noise into shape boundaries described by Freeman chain codes. Shapes retain their property of being closed or open after noise is injected into their chain code sequences due to the design of the proposed method. Furthermore, during the noise injection, the procedure prevents alterations that would cause changes in shape’s topological characteristics. Different test shapes were used to demonstrate noise injection into chain codes. Although the difference of their properties is immense, the method managed to process all shapes successfully regardless of their complexity. The evaluation metrics fractal dimension and mean distance from original were used for estimating the degree of injected noise in a shape. The proposed method provides an efficient process of noise injection performed at the chain code level. To the best of our knowledge, it is the first method that enables direct noise injection into chain codes without the need to rasterise and re-encode the geometric shapes.

Noise injection into chain codes could serve as a preprocessing phase of different methods by producing noisy datasets. Especially in the field of deep neural networks, predominantly dealing with chain codes (Sanchiz, Iñesta & Pla, 1996; Subri, Haron & Sallehuddin, 2006), adding noise to the training data could be a crucial step in order to reduce the influence of phenomena that occur during the training phase of a neural network. Therefore, data augmentation in the form of providing noisy datasets would be of the utmost importance for enhancing the performance of neural networks.

There are several directions for future work. The most obvious improvement would be to adapt the method to process other types of chain codes, such as VCC, 3OT, UMCC, chain codes based on biological computing, or multi-resolution chain codes. Apart from the encoding type, an adjustment could be made to the method in order to enable the noise injection into 3D chain codes (Sánchez-Cruz, López-Valdez & Cuevas, 2014; Strnad et al., 2019). Furthermore, the method could be applied to machine learning (ML), where the performance of ML models could be evaluated under varying degrees of noise injected into their datasets.

Additional Information and Declarations

Competing Interests

The authors have a patent with the number EP24150479.4 pending to University of Maribor.

Author Contributions

Luka Lukač conceived and designed the experiments, performed the experiments, analyzed the data, performed the computation work, prepared figures and/or tables, and approved the final draft.

Andrej Nerat conceived and designed the experiments, performed the experiments, prepared figures and/or tables, authored or reviewed drafts of the article, and approved the final draft.

Damjan Strnad analyzed the data, performed the computation work, prepared figures and/or tables, authored or reviewed drafts of the article, and approved the final draft.

Ivana Kolingerová analyzed the data, authored or reviewed drafts of the article, and approved the final draft.

Borut Žalik conceived and designed the experiments, prepared figures and/or tables, authored or reviewed drafts of the article, and approved the final draft.

Patent Disclosures

The following patent dependencies were disclosed by the authors:

Patent application number: EP24150479.4

Name: A method for noise injection into 2D geometric shapes described by Freeman chain codes

Status: pending

Organization: University of Maribor

Authors: Luka Lukač, Damjan Strnad, Andrej Nerat, Borut Žalik

Date of filing: 5. 1. 2024

Data Availability

The following information was supplied regarding data availability:

The dataset that contains the testing examples is available at: https://gemma.feri.um.si/chaincodes/ (accessed on 13 February 2025).

The source code of the method is available at Github and Zenodo:

- https://github.com/luckyLukac/FreemanChainCodeNoiseInjection.

- luckyLukac. (2025). luckyLukac/FreemanChainCodeNoiseInjection: Version 1.00 (release). Zenodo. https://doi.org/10.5281/zenodo.15877994.

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
