# Peer review of "Noise injection into Freeman chain codes"

_PeerJ Computer Science, doi:10.7717/peerj-cs.3084_

## Round 0.1 · original submission · Major Revisions

The reviewers have provided extensive feedback on the paper, with suggestions to improve its quality and clarity. The main concerns include the need for a more detailed literature review, discussion of previous solutions, and justification of certain choices made in the methodology.

The reviews emphasize the need for a formal mathematical description or proof-style discussion of the proposed method, especially regarding the logic for preserving self-intersections and holes. They also recommend including a comparison with indirect noise injection methods and discussing the advantages and disadvantages of using traditional Freeman chain codes versus modern alternatives.

They also provide suggestions to enhance the paper's impact and reproducibility, including justifying the choice of dataset, discussing alternative metrics for quantifying noise levels, and considering time and space complexity analysis. They also suggest including a discussion on biological computing approaches in chain coding, which has emerged as a new trend in the field.

One reviewer highlights the importance of addressing limitations and providing future work directions in the conclusions. They also suggest improving the presentation of results and adding more systematic quantitative and qualitative measurements.

Reviewer 1 ·

Basic reporting

In this paper, the authors introduce the first method for injecting noise directly into geometric shapes represented by Freeman chain codes without requiring rasterization and re-encoding. The proposed technique applies predefined alterations while preserving key topological characteristics. Its effectiveness is validated through experiments on diverse shapes, with Fractal Dimension used to quantify noise levels. I have some revisions to be addressed before I can accept the paper for publication:
(1) For your research, you used the GeMMA dataset. Is this a standardized dataset that is used by researchers in your community? What is the advantage of using this dataset as opposed to others in the field? Justify the usage of the dataset.
(2) You used the Fractal Dimension (FD), which seems to be used to quantify the degree of injected noise. Are there other metrics in the field? I need a justification for why you went with FD.
(3) The logic for preserving self-intersections and holes would benefit from more mathematical formalism or proof-style discussion.
(4) The literature review is not current. For example, a new trend in chain coding employs biological computing and the behavior of some creatures to encode images.
Some papers utilize dolphins, tigers, and predator-prey relationships. I don’t see that you included that literature.
(5) Including a comparison with indirect noise injection (e.g., rasterizing, adding pixel-level noise, then re-encoding) would highlight the benefits of your direct approach more clearly in terms of fidelity and efficiency.
(6) Justify the usage of the traditional Freeman chain code in your research. Why did you not use a modern chain code, such as the one that uses solitary behavior in tigers to address the problem? Do you think using modern code will generate better results?
(7) You can briefly discuss the time and space complexity of the noise injection algorithm.
(8) You can consider having a more formal mathematical description or a visual proof framework to strengthen the mechanism you are providing in this manuscript.

Experimental design

The experimental design is well-structured and appropriate for validating the proposed method. I appreciate that the method can be reproduced.

Validity of the findings

The findings are valid and well-supported by both qualitative visual results and quantitative analysis using Fractal Dimension.

Cite this review as
Anonymous Reviewer (2025) Peer Review #1 of "Noise injection into Freeman chain codes (v0.1)". PeerJ Computer Science

·

Basic reporting

Comment 1. The introduction section needs to be updated with more relevant research. In this sense, a more detailed review of the literature is expected. Also, it is required that the previous solutions to this problem be addressed. Then, the proposed method's advantages (and disadvantages) should be discussed. You must update the reference list to contain articles from at least five years ago.

Comment 2. The motivation and the contribution of the proposed method must be discussed in the Introduction section. Please provide more details to understand them better.

Experimental design

Comment 3. The authors should do a better job of commenting on the results. More comparisons of results and explanation of the advantages of the eight- or four-directional methods over other proposed techniques are needed.

Comment 4. What are the limitations of the methodology adopted in this research?

Comment 5. How was the value μ = 0.1 chosen to inject noise? Describe the justification.

Validity of the findings

Comment 7. In the Conclusions, there is insufficient discussion of exactly what this finding means and its implications. In addition, they should be enriched with future work.

Additional comments

The paper presents the first method for direct noise injection into geometric shapes described by eight- or four-directional Freeman chain codes. The topic is interesting and timely and has potential applicability. However, the paper needs some corrections.

Cite this review as

·

Basic reporting

Please add this information:
1. There is no discussion on the Injecting noise that can disrupt the continuity of chain codes, making it harder to reconstruct the original shape.
2. The complexity level of the proposed method?

Experimental design

1. Small perturbations in the chain code can lead to significant deviations in shape representation, affecting applications like pattern recognition. How the proposed method can handle this?
2. Poor presentation of the results. Please add more systematic results and discussion at every stage. These must be in qualitative and quantitative measurements.
3. More technical explanation on Figure 22.

Validity of the findings

It depends on the experimental design to validate the finding. Please also do some comparison with state-of-the-art methods.

Additional comments

Please also discuss the reconstruction accuracy and filtering effectiveness.

Cite this review as

---

## Round 0.2 · accepted · Accept

The noise injection method is an elegant strategy that supports the creation of training datasets for neuronal network algorithms.

I only would recommend the authors to review their abstract: I suggest eliminating "the first" method (such bold claims are usually not well perceived; experts will recognize this anyway). In addition, the authors should include at the end of the abstract their main scientific contribution and possible applications.

Reviewer 1 ·

Basic reporting

The author addressed the revisions. I am fine with accepting the paper

Experimental design

The author addressed the revisions. I am fine with accepting the paper

Validity of the findings

The author addressed the revisions. I am fine with accepting the paper

Additional comments

The author addressed the revisions. I am fine with accepting the paper

Cite this review as
Anonymous Reviewer (2025) Peer Review #1 of "Noise injection into Freeman chain codes (v0.2)". PeerJ Computer Science

·

Basic reporting

The introduction provides appropriate background and clearly describes the motivation for the research, emphasizing the novelty of directly injecting noise into Freeman chain codes.

The literature review is relevant, up-to-date, and sufficiently covers both theoretical and applied aspects of chain codes in data compression and machine learning.

The manuscript structure conforms to PeerJ’s standards, with logically organized sections including Introduction, Methods, Results, and Conclusions.

Experimental design

The study presents original research well within the scope of the journal.

The specific problem addressed by this study represents a gap in the existing body of literature, i.e., the lack of methodologies for performing direct noise injection on chain code representations of geometric shapes.

The authors test their method on a diverse set of shapes, varying in complexity and number of loops/rings, which strengthens the generalizability of their findings.

Validity of the findings

The findings are coherent and well-supported by both quantitative metrics and visual results. The preservation of topological characteristics is consistently maintained across various shapes and noise levels. Moreover, the use of Fractal Dimension and Mean Distance from Original provides reliable indicators to assess noise injection effects.

Cite this review as